# Quantum–Classical Correspondence Principle for Heat Distribution in Quantum Brownian Motion

**DOI:** 10.3390/e23121602

**Published:** 2021-11-26

**Authors:** Jin-Fu Chen, Tian Qiu, Hai-Tao Quan

**Affiliations:** 1School of Physics, Peking University, Beijing 100871, China; chenjinfu@pku.edu.cn (J.-F.C.); tianqiu2016@pku.edu.cn (T.Q.); 2Collaborative Innovation Center of Quantum Matter, Beijing 100871, China; 3Frontiers Science Center for Nano-Optoelectronics, Peking University, Beijing 100871, China

**Keywords:** open quantum systems, phase-space formulation, quantum Brownian motion, heat statistics

## Abstract

Quantum Brownian motion, described by the Caldeira–Leggett model, brings insights to the understanding of phenomena and essence of quantum thermodynamics, especially the quantum work and heat associated with their classical counterparts. By employing the phase-space formulation approach, we study the heat distribution of a relaxation process in the quantum Brownian motion model. The analytical result of the characteristic function of heat is obtained at any relaxation time with an arbitrary friction coefficient. By taking the classical limit, such a result approaches the heat distribution of the classical Brownian motion described by the Langevin equation, indicating the quantum–classical correspondence principle for heat distribution. We also demonstrate that the fluctuating heat at any relaxation time satisfies the exchange fluctuation theorem of heat and its long-time limit reflects the complete thermalization of the system. Our research study justifies the definition of the quantum fluctuating heat via two-point measurements.

## 1. Introduction

In the past few decades, the discovery of fluctuation theorems [1,2,3,4] and the establishment of the framework of stochastic thermodynamics [5,6,7] deepened our understanding of the fluctuating nature of thermodynamic quantities (such as work, heat and entropy production) in microscopic systems [8,9,10,11,12,13]. Among various fluctuation theorems, the non-equilibrium work relation [2] sharpens our understanding of the second law of thermodynamics by presenting an elegant and precise equality associating the free energy change with the fluctuating work. Such a relation was later extended to the quantum realm based on the two-point measurement definition of the quantum fluctuating work [14,15], soon after its discovery in the classical regime. The work statistics has been widely studied in various microscopic classical and quantum systems [16,17,18,19,20,21,22,23,24,25,26]. Historically, the quantum–classical correspondence principle played an essential role in the development of the theory of quantum mechanics and the interpretation of the transition from quantum to classical world [27,28]. In Refs. [19,22,24], it is demonstrated that the existence of the quantum–classical correspondence principle for work distribution brings justification for the definition of quantum fluctuating work via two-point measurements.

Compared to work statistics, heat statistics relevant to thermal transport associated with a nonequilibrium stationary state has been extensively studied [29,30,31,32,33,34,35,36,37,38], but the heat statistics in a finite-time quantum thermodynamic process [39,40,41] and its quantum–classical correspondence have been less explored. A challenge is that the precise description of the bath dynamics requires handling a huge number of degrees of freedom of the heat bath. Different approaches have been proposed to calculate the quantum fluctuating heat and its statistics, such as the non-equilibrium Green’s function approach to quantum thermal transport [29,32,36,42,43,44] and the path-integral approach to quantum thermodynamics [45,46,47,48,49]. However, very few analytical results about the heat statistics have been obtained for the relaxation processes in open quantum systems. These analytical results are limited to either the relaxation dynamics described by the Lindblad master equation [39,40] or the long-time limit independent of the relaxation dynamics [50]. On the other hand, some results about the heat statistics in the classical Brownian motion model have been reported [51,52,53,54,55,56,57,58,59]. How the quantum and the classical heat statistics (especially associated with the relaxation dynamics in finite time) are related to each other has not been explored so far, probably due to the difficulty in studying the heat statistics in open quantum systems [60,61,62].

In this article, we study the heat statistics of a quantum Brownian motion model described by the Caldeira–Leggett Hamiltonian [48,63,64,65,66,67,68], where the heat bath is modeled as a collection of harmonic oscillators. Although it is well known that the dynamics of such an open quantum system can approach that of the classical Brownian motion in the classical limit 
ℏ→0
 [64], less is known about the heat statistics of this model during the finite-time relaxation process. Here, we focus on the relaxation process without external driving (the Hamiltonian of the system is time-independent); thus, the quantum fluctuating heat can be defined as the difference of the system energy between the initial and the final measurements [69].Under the Ohmic spectral density, the dynamics of the composite system is exactly solvable in the continuum limit of the bath oscillators [70]. By employing the phase-space formulation approach [71,72,73], we obtain analytical results of the characteristic function of heat for the Caldeira–Leggett model at any relaxation time 
τ
 with an arbitrary friction coefficient 
κ
. Previously, such an approach was employed to study the quantum corrections to work [74,75,76] and entropy [77,78]. Analytical results of the heat statistics bring important insights to understand the fluctuating property of heat. By taking the classical limit 
ℏ→0
, the heat statistics of the Caldeira–Leggett model approaches that of the classical Brownian motion model. Thus, our results verify the quantum–classical correspondence principle for heat distribution, and provide justification for the definition of the quantum fluctuating heat via two-point measurements. We also verify, from the analytical results, that the heat statistics satisfies the exchange fluctuation theorem of heat [4].

The rest of this article is organized as follows. In Section 2, we introduce the Caldeira–Leggett model and define the quantum fluctuating heat. In Section 3, the analytical results of the characteristic function of heat are obtained by employing the phase-space formulation approach. We show the quantum–classical correspondence of the heat distribution and discuss the heat distribution in the long-time limit or with the extremely weak or strong coupling strength. The conclusion is given in Section 4.

## 2. The Caldeira–Leggett Model and the Heat Statistics

### 2.1. The Caldeira–Leggett Model

The quantum Brownian motion is generally described by the Caldeira–Leggett model [64,65], where the system is modeled as a single particle moving in a specific potential and the heat bath is a collection of harmonic oscillators. For simplicity, we choose the harmonic potential for the system [66,79,80,81], where the dynamics of such an open quantum system can be solved analytically. The system relaxes to the equilibrium state at the temperature of the heat bath. We study the heat distribution of such a quantum relaxation process and analytically obtain the characteristic function of heat and its classical correspondence based on the phase-space formulation of quantum mechanics.

The total Hamiltonian of the composite system is 
$H_{\mathrm{tot}}=H_{S}+H_{B}+H_{SB}$
 with each term being

(1)
$$\begin{array}{cc} H_{S} & =\frac{1}{2}\frac{{\hat{p}}_{0}^{2}}{m_{0}}+\frac{1}{2}m_{0}\omega_{0}^{2}{\hat{q}}_{0}^{2} \end{array}$$


(2)
$$\begin{array}{cc} H_{B} & =\sum_{n=1}^{N}\left(\frac{1}{2}\frac{{\hat{p}}_{n}^{2}}{m_{n}}+\frac{1}{2}m_{n}\omega_{n}^{2}{\hat{q}}_{n}^{2}\right) \end{array}$$


(3)
$$\begin{array}{cc} H_{SB} & =-{\hat{q}}_{0}\sum_{n=1}^{N}\left(C_{n}{\hat{q}}_{n}\right)+\sum_{n=1}^{N}\left(\frac{C_{n}^{2}}{2m_{n}\omega_{n}^{2}}{\hat{q}}_{0}^{2}\right), \end{array}$$

where 
$m_{0}$
, 
$\omega_{0}$
, 
${\hat{q}}_{0}$
 and 
${\hat{p}}_{0}$
 (
$m_{n}$
, 
$\omega_{n}$
, 
${\hat{q}}_{n}$
 and 
${\hat{p}}_{n}$
 with 
n=1,2,3,...,N
) are the mass, frequency, position and momentum of the system (the *n*-th bath harmonic oscillator) and 
$C_{n}$
 is the coupling strength between the system and the *n*-th bath harmonic oscillator. The counter-term 
$\sum_{n}\left[C_{n}^{2}/\left(2m_{n}\omega_{n}^{2}\right)\right]{\hat{q}}_{0}^{2}$
 is included in the interaction Hamiltonian 
$H_{SB}$
 to cancel the frequency shift of the system.

The spectral density is defined as 
$J\left(\omega \right):=\sum_{n}\left[C_{n}^{2}/\left(2m_{n}\omega_{n}\right)\right]\delta \left(\omega -\omega_{n}\right)$
. We adopt an Ohmic spectral density with the Lorentz–Drude cutoff [67]

(4)
$$J\left(\omega \right)=\frac{m_{0}\kappa }{\pi }\omega \frac{\Omega_{0}^{2}}{\Omega_{0}^{2}+\omega^{2}},$$

where 
κ
 is the friction coefficient. A sufficiently large cutoff frequency 
$\Omega_{0}$
 (
$\Omega_{0}\gg \omega_{0}$
) is applied to ensure a finite counter-term and the dynamics with the timescale exceeding 
$1/\Omega_{0}$
 is Markovian. Under such a spectral density, the dissipation dynamics of the Caldeira–Leggett model with a weak coupling strength 
$\kappa \ll \omega_{0}$
 reproduces that of the classical underdamped Brownian motion when taking the classical limit 
ℏ→0
 [64].

We assume the initial state to be a product state of the system and the heat bath

(5)
$$\rho \left(0\right)=\rho_{S}\left(0\right)⊗\rho_{B}^{G},$$

which makes it possible to define the quantum fluctuating heat via two-point measurements. Here, 
$\rho_{S}\left(0\right)$
 is the initial state of the system and 
$\rho_{B}^{G}=\exp\left(-\beta H_{B}\right)/Z_{B}\left(\beta \right)$
 is the Gibbs distribution of the heat bath with the inverse temperature 
β
 and the partition function 
$Z_{B}\left(\beta \right)=\mathrm{Tr}\left[\exp\left(-\beta H_{B}\right)\right]$
.

### 2.2. The Quantum Fluctuating Heat in the Relaxation Process

We study the heat distribution of the relaxation process based on the two-point measurement definition of the quantum fluctuating heat. When no external driving is applied to the system, the Hamiltonian of the system is time-independent. Since no work is performed during the relaxation process, the quantum fluctuating heat can be defined as

(6)
$$Q_{l^{\prime }l}=E_{l^{\prime }}^{S}-E_{l}^{S},$$

where 
$E_{l}^{S}$
 (
$E_{l^{\prime }}^{S}$
) is the eigenenergy of the system corresponding to the outcome *l* (
$l^{\prime }$
) at the initial (final) time 
t=0
 (
t=τ
). The two-point measurements over the heat bath can be hardly realized due to a huge number of degrees of freedom of the heat bath [20], while the measurements over the small quantum system are much easier in principle. The positive sign corresponds to the energy flowing from the heat bath to the system.

For the system prepared in an equilibrium state, no coherence exists in the initial state and the initial density matrix of the system commutes with the Hamiltonian of the system, 
$[\rho \left(0\right),H_{S}]=0$
. The probability of observing the transition from *l* and 
$l^{\prime }$
 is

(7)
$$p_{\tau ,l^{\prime }l}=\gamma_{\tau ,l^{\prime }l}p_{l},$$

with the conditional transition probability 
$\gamma_{\tau ,l^{\prime }l}=\mathrm{Tr}\left[\left({\hat{P}}_{l^{\prime }}^{S}⊗I_{B}\right)U_{\mathrm{tot}}\left(\tau \right)\left({\hat{P}}_{l}^{S}⊗\rho_{B}^{G}\right)U_{\mathrm{tot}}^{†}\left(\tau \right)\right]$
 and the initial probability 
$p_{l}=\mathrm{Tr}\left[\rho \left(0\right){\hat{P}}_{l}^{S}\right]$
. Here, 
${\hat{P}}_{l}^{S}=|l\rangle \langle l|$
 is the projection operator corresponding to the outcome *l*. The heat distribution is defined as

(8)
$$P_{\tau }\left(q\right):=\sum_{l^{\prime },l}\delta \left(q-Q_{l^{\prime }l}\right)p_{\tau ,l^{\prime }l}.$$


The characteristic function of heat 
$\chi_{\tau }\left(\nu \right)$
 is defined as the Fourier transform of the heat distribution 
$\chi_{\tau }\left(\nu \right):=\sum_{l^{\prime },l}\exp\left[i\nu \left(E_{l^{\prime }}^{S}-E_{l}^{S}\right)\right]p_{\tau ,l^{\prime }l}$
, which can be rewritten explicitly as

(9)
$$\chi {}_{\tau }\left(\nu \right)=\mathrm{Tr}\left[e^{i\nu H_{S}}U_{\mathrm{tot}}\left(\tau \right)\left(e^{-i\nu H_{S}}\rho \left(0\right)\right)U_{\mathrm{tot}}^{†}\left(\tau \right)\right],$$

where 
$U_{\mathrm{tot}}\left(\tau \right)=\exp\left(-iH_{\mathrm{tot}}\tau /\hbar \right)$
 is the unitary time-evolution operator of the composite system.

Our goal is to analytically calculate the characteristic function 
$\chi {}_{\tau }\left(\nu \right)$
. Previously, the quantum–classical correspondence principle for heat statistics has been analyzed with the path-integral approach to quantum thermodynamics [48], yet the explicit result of the characteristic function (or generating function) of heat has not been obtained so far. We employ the phase-space formulation approach to solve this problem and rewrite the characteristic function Equation (Equation 9) into

(10)
$$\chi {}_{\tau }\left(\nu \right)=\mathrm{Tr}\left[e^{i\nu H_{S}^{H}\left(\tau \right)}\eta \left(0\right)\right],$$

where the system Hamiltonian in the Heisenberg picture is

(11)
$$H_{S}^{H}\left(\tau \right)=U_{\mathrm{tot}}^{†}\left(\tau \right)H_{S}U_{\mathrm{tot}}\left(\tau \right),$$

and the density matrix-like operator 
η(0)
 is

(12)
$$\eta \left(0\right)=\left[e^{-i\nu H_{S}}\rho_{S}\left(0\right)\right]⊗\rho_{B}^{G}.$$


We express Equation (Equation 10) with the phase-space formulation of quantum mechanics [71,72,73,74,75,76]:
(13)
$$\chi_{\tau }\left(\nu \right)=\frac{1}{{\left(2\pi \hbar \right)}^{N+1}}\int dz{\left[e^{i\nu H_{S}^{H}\left(\tau \right)}\right]}_{w}\left(z\right)\cdot P\left(z\right),$$

where 
z
 represents a point 
$z=\left[q,p\right]=\left[q_{0},...,q_{N},p_{0},...,p_{N}\right]$
 in the phase space of the composite system and the integral is performed over the whole phase space. The subscript “*w*” indicates the Weyl symbol of the corresponding operator and 
P(z)
 is the Weyl symbol of the operator 
η(0)
, which is explicitly defined as [71]

(14)
$$P\left(z\right):=\int dy\langle q-\frac{y}{2}|\eta \left(0\right)|q+\frac{y}{2}\rangle e^{\frac{ip\cdot y}{\hbar }}.$$


In the following, we calculate the heat statistics Equation (Equation 13) by employing the phase-space formulation approach.

## 3. Results of the Characteristic Function of Heat

We show a sketch of the derivation of the heat statistics 
$\chi_{\tau }\left(\nu \right)$
 with the details left in Appendix A. We specifically consider the system is initially prepared at an equilibrium state 
$\rho_{S}\left(0\right)=\exp\left(-\beta^{\prime }H_{S}\right)/Z_{S}\left(\beta^{\prime }\right)$
 with the inverse temperature 
$\beta^{\prime }$
 and the partition function 
$Z_{S}\left(\beta^{\prime }\right)=1/\left[2\sinh\left(\beta^{\prime }\hbar \omega_{0}/2\right)\right]$
. The heat bath is at the inverse temperature 
β
, which is different from 
$\beta^{\prime }$
. In Equation (Equation 13), the two Weyl symbols 
${\left[e^{i\nu H_{S}^{H}\left(\tau \right)}\right]}_{w}\left(z\right)$
 and 
P(z)
 are obtained as

(15)
$${\left[e^{i\nu H_{S}^{H}\left(\tau \right)}\right]}_{w}\left(z\right)=\frac{1}{\cos\left(\frac{\nu \hbar \omega_{0}}{2}\right)}\exp\left[\frac{i}{2\hbar }z^{T}{\tilde{\Lambda }}_{\nu z}\left(\tau \right)z\right],$$

and

(16)
$$P\left(z\right)=\frac{2\sinh\left(\frac{\beta^{\prime }\hbar \omega_{0}}{2}\right)}{\cosh\left[\frac{\left(\beta^{\prime }+i\nu \right)\hbar \omega_{0}}{2}\right]}\cdot \left[\prod_{n=1}^{N}2\tanh\left(\frac{\beta \hbar \omega_{n}}{2}\right)\right]\cdot \exp\left(-\frac{1}{2\hbar }z^{T}\Lambda_{\beta z}z\right),$$

where the explicit expressions of the matrices 
${\tilde{\Lambda }}_{\nu z}\left(\tau \right)$
 and 
$\Lambda_{\beta z}$
 are given in Equations (Equation 48) and (Equation 74), respectively.

Substituting Equations (Equation 15) and (Equation 16) into Equation (Equation 13), the characteristic function of heat at any relaxation time 
τ
 with an arbitrary friction coefficient 
κ
 is finally obtained as

(17)
$$\begin{array}{cc} \chi_{\tau }\left(\nu \right)= & \left\{{\left[\left(1+i\Xi \right)\left(1-i\Theta \Xi \right)-i\Xi \left(1-\Theta -i\Theta \Xi \right)\frac{\kappa^{2}\cos\left(2{\hat{\omega }}_{0}\tau \right)-4\omega_{0}^{2}}{\left(\kappa^{2}-4\omega_{0}^{2}\right)e^{\kappa \tau }}\right]}^{2}\right.\\ \; & {\left.+\Xi^{2}{\left(1-\Theta -i\Theta \Xi \right)}^{2}\left[{\left(\frac{\kappa^{2}\cos\left(2{\hat{\omega }}_{0}\tau \right)-4\omega_{0}^{2}}{\left(\kappa^{2}-4\omega_{0}^{2}\right)e^{\kappa \tau }}\right)}^{2}-e^{-2\kappa \tau }\right]\right\}}^{\frac{1}{2}}, \end{array}$$

where the quantities 
Ξ
 and 
Θ
 are

(18)
$$\begin{array}{cc} \Xi & =\frac{\tan\left(\frac{\nu \hbar \omega_{0}}{2}\right)}{\tanh\left[\frac{\left(\beta^{\prime }+i\nu \right)\hbar \omega_{0}}{2}\right]-i\tan\left(\frac{\nu \hbar \omega_{0}}{2}\right)}, \end{array}$$


(19)
$$\begin{array}{cc} \Theta & =\frac{\tanh\left[\frac{\left(\beta^{\prime }+i\nu \right)\hbar \omega_{0}}{2}\right]-i\tan\left(\frac{\nu \hbar \omega_{0}}{2}\right)}{\tanh\left(\frac{\beta \hbar \omega_{0}}{2}\right)}. \end{array}$$


Induced by the friction, the frequency of the system harmonic oscillator is shifted to 
${\hat{\omega }}_{0}=\sqrt{\omega_{0}^{2}-\kappa^{2}/4}$
.

From the analytical results of the heat statistics Equation (Equation 17), the average heat 
$\langle Q\rangle \left(\tau \right)={\left.-i\partial_{\nu }\left[\ln\chi_{\tau }\left(\nu \right)\right]\right|}_{\nu =0}$
 is immediately obtained as

(20)
$$\begin{array}{cc} \langle Q\rangle \left(\tau \right) & =\frac{\omega_{0}\hbar }{2}\left[\coth\left(\frac{\beta \omega_{0}\hbar }{2}\right)-\coth\left(\frac{\beta^{\prime }\omega_{0}\hbar }{2}\right)\right]\left[1-\frac{\kappa^{2}\cos\left(2{\hat{\omega }}_{0}\tau \right)-4\omega_{0}^{2}}{\left(\kappa^{2}-4\omega_{0}^{2}\right)e^{\kappa \tau }}\right], \end{array}$$

and the variance 
$\mathrm{Var}\left(Q\right)\left(\tau \right)={\left.-\partial_{\nu }^{2}\left[\ln\chi_{\tau }\left(\nu \right)\right]\right|}_{\nu =0}$
 is

(21)
$$\mathrm{Var}\left(Q\right)\left(\tau \right)=I+\mathrm{II}\cdot e^{-\kappa \tau }+\mathrm{III}\cdot e^{-2\kappa \tau },$$

with

(22)
$$\begin{array}{cc} I & =\frac{\omega_{0}^{2}\hbar^{2}\left[{\mathrm{csch}}^{2}\left(\frac{\beta \omega_{0}\hbar }{2}\right)+{\mathrm{csch}}^{2}\left(\frac{\beta^{\prime }\omega_{0}\hbar }{2}\right)\right]}{4}, \end{array}$$


(23)
$$\begin{array}{cc} \mathrm{II} & =\frac{\kappa^{2}\cos\left(2{\hat{\omega }}_{0}\tau \right)-4\omega_{0}^{2}}{2{\hat{\omega }}_{0}^{2}}\cdot \frac{\omega_{0}^{2}\hbar^{2}\left[\coth^{2}\left(\frac{\beta \omega_{0}\hbar }{2}\right)+{\mathrm{csch}}^{2}\left(\frac{\beta^{\prime }\omega_{0}\hbar }{2}\right)-\coth\left(\frac{\beta \omega_{0}\hbar }{2}\right)\coth\left(\frac{\beta^{\prime }\omega_{0}\hbar }{2}\right)\right]}{4} \end{array}$$


(24)
$$\begin{array}{cc} \mathrm{III} & =\frac{\kappa^{4}\cos\left(4{\hat{\omega }}_{0}\tau \right)+8\omega_{0}^{2}\kappa^{2}\left[1-2\cos\left(2{\hat{\omega }}_{0}\tau \right)\right]+16\omega_{0}^{4}}{16{\hat{\omega }}_{0}^{4}}\cdot \frac{\omega_{0}^{2}\hbar^{2}{\left[\coth\left(\frac{\beta \omega_{0}\hbar }{2}\right)-\coth\left(\frac{\beta^{\prime }\omega_{0}\hbar }{2}\right)\right]}^{2}}{4}. \end{array}$$


Similarly, one can calculate the higher cumulants from the analytical results of the heat statistics. In the following, we examine the properties of the heat statistics of the quantum Brownian motion.

### 3.1. Quantum–Classical Correspondence Principle for Heat Statics and the
Exchange Fluctuation Theorem of Heat

We further take the classical limit 
ℏ→0
 or, more rigorously, 
$\beta \hbar \omega_{0}\to 0$
. The two quantities approach 
$\Xi \to \nu /\beta^{\prime }$
 and 
$\Theta \to \beta^{\prime }/\beta$
 and the characteristic function of heat (Equation (Equation 17)) becomes

(25)
$$\begin{array}{cc} \chi_{\tau }^{\mathrm{cl}}\left(\nu \right)= & \left\{{\left[\left(1+i\frac{\nu }{\beta^{\prime }}\right)\left(1-i\frac{\nu }{\beta }\right)-\frac{i\nu \left(\beta -\beta^{\prime }-i\nu \right)}{\beta \beta^{\prime }}\frac{\left(\kappa^{2}\cos\left(2{\hat{\omega }}_{0}\tau \right)-4\omega_{0}^{2}\right)}{\left(\kappa^{2}-4\omega_{0}^{2}\right)e^{\kappa \tau }}\right]}^{2}\right.\\ \; & {\left.+\nu^{2}{\left(\frac{\beta -\beta^{\prime }-i\nu }{\beta \beta^{\prime }}\right)}^{2}\left[{\left(\frac{\kappa^{2}\cos\left(2{\hat{\omega }}_{0}\tau \right)-4\omega_{0}^{2}}{\left(\kappa^{2}-4\omega_{0}^{2}\right)e^{\kappa \tau }}\right)}^{2}-e^{-2\kappa \tau }\right]\right\}}^{-\frac{1}{2}}, \end{array}$$

which is consistent with the results obtained from the classical Brownian motion described by the Kramers equation (see Ref. [58] or Appendix C). The average heat is

(26)
$$\begin{array}{cc} \langle Q^{\mathrm{cl}}\rangle \left(\tau \right) & =\frac{\beta^{\prime }-\beta }{\beta \beta^{\prime }}\left[1-\frac{\kappa^{2}\cos\left(2{\hat{\omega }}_{0}\tau \right)-4\omega_{0}^{2}}{\left(\kappa^{2}-4\omega_{0}^{2}\right)e^{\kappa \tau }}\right], \end{array}$$

and the variance 
$\mathrm{Var}\left(Q^{\mathrm{cl}}\right)\left(\tau \right)={\left.-\partial_{\nu }^{2}\left[\ln\chi_{\tau }^{\mathrm{cl}}\left(\nu \right)\right]\right|}_{\nu =0}$
 is

(27)
$$\begin{array}{cc} \mathrm{Var}\left(Q^{\mathrm{cl}}\right)\left(\tau \right) & =I^{\mathrm{cl}}+{\mathrm{II}}^{\mathrm{cl}}\cdot e^{-\kappa \tau }+{\mathrm{III}}^{\mathrm{cl}}\cdot e^{-2\kappa \tau }, \end{array}$$

with

(28)
$$\begin{array}{cc} I^{\mathrm{cl}} & =\frac{\beta^{2}+\beta^{\prime 2}}{\beta^{2}\beta^{\prime 2}} \end{array}$$


(29)
$$\begin{array}{cc} {\mathrm{II}}^{\mathrm{cl}} & =\frac{\kappa^{2}\cos\left(2{\hat{\omega }}_{0}\tau \right)-4\omega_{0}^{2}}{2{\hat{\omega }}_{0}^{2}}\cdot \frac{\beta^{2}-\beta \beta^{\prime }+\beta^{\prime 2}}{\beta^{2}\beta^{\prime 2}} \end{array}$$


(30)
$$\begin{array}{cc} {\mathrm{III}}^{\mathrm{cl}} & =\frac{\kappa^{4}\cos\left(4{\hat{\omega }}_{0}\tau \right)+8\omega_{0}^{2}\kappa^{2}\left[1-2\cos\left(2{\hat{\omega }}_{0}\tau \right)\right]+16\omega_{0}^{4}}{16{\hat{\omega }}_{0}^{4}}\cdot \frac{{\left(\beta -\beta^{\prime }\right)}^{2}}{\beta^{2}\beta^{\prime 2}}. \end{array}$$


From Equation (Equation 17) (or the classical counterpart Equation (Equation 25)), one can see the characteristic function of heat exhibits the following symmetry:
(31)
$$\chi_{\tau }\left(\nu \right)=\chi_{\tau }\left[-i\left(\beta -\beta^{\prime }\right)-\nu \right],$$

which shows that the heat distribution satisfies the exchange fluctuation theorem of heat in the differential form 
$P_{\tau }\left(Q\right)/P_{\tau }\left(-Q\right)=\exp\left[-\left(\beta -\beta^{\prime }\right)Q\right]$
 [4]. By setting 
ν=0
, we obtain the relation 
$\chi_{\tau }\left[-i\left(\beta -\beta^{\prime }\right)\right]=\chi_{\tau }\left(0\right)=1$
, which is exactly the exchange fluctuation theorem of heat in the integral form 
$\langle \exp[\left(\beta -\beta^{\prime }\right)Q]\rangle =1$
.

### 3.2. Long-Time Limit

In the long-time limit 
τ→∞
, the characteristic functions of heat (Equations (Equation 17) and (Equation 25)) become

(32)
$$\begin{array}{cc} \chi_{\infty }\left(\nu \right) & =\frac{\left(1-e^{-\beta^{\prime }\omega_{0}\hbar }\right)\left(1-e^{-\beta \omega_{0}\hbar }\right)}{\left(1-e^{-\left(\beta^{\prime }+i\nu \right)\omega_{0}\hbar }\right)\left(1-e^{-\left(\beta -i\nu \right)\omega_{0}\hbar }\right)}, \end{array}$$

and

(33)
$$\chi_{\infty }^{\mathrm{cl}}\left(\nu \right)=\frac{\beta^{\prime }\beta }{\left(\beta^{\prime }+i\nu \right)\left(\beta -i\nu \right)}.$$


Such results, independent of the relaxation dynamics, are in the form

(34)
$$\chi_{\mathrm{th}}\left(\nu \right)=\frac{Z_{S}\left(\beta^{\prime }+i\nu \right)Z_{S}\left(\beta -i\nu \right)}{Z_{S}\left(\beta^{\prime }\right)Z_{S}\left(\beta \right)},$$

reflecting complete thermalization of the system [53]. For example, the relaxation of a harmonic oscillator governed by the quantum–optical master equation gives the identical characteristic function of heat in the long-time limit [39]. In Appendix D, we demonstrate that the characteristic function of heat for any relaxation process with complete thermalization is always in the form of Equation (Equation 34). With the simple expressions (Equation 32) and (Equation 33) of the characteristic functions, the heat distributions are obtained from the inverse Fourier transform as

(35)
$$P_{\infty }\left(q\right)=\left\{\begin{array}{cc} \frac{\left(1-e^{-\beta^{\prime }\omega_{0}\hbar }\right)\left(1-e^{-\beta \omega_{0}\hbar }\right)}{1-e^{-\left(\beta^{\prime }+\beta \right)\omega_{0}\hbar }}\sum_{j=0}^{\infty }\delta \left(q-j\omega_{0}\hbar \right)e^{-\beta q} & q\geq 0\\ \frac{\left(1-e^{-\beta^{\prime }\omega_{0}\hbar }\right)\left(1-e^{-\beta \omega_{0}\hbar }\right)}{1-e^{-\left(\beta^{\prime }+\beta \right)\omega_{0}\hbar }}\sum_{j=1}^{\infty }\delta \left(q+j\omega_{0}\hbar \right)e^{\beta^{\prime }q} & q<0 \end{array}\right.,$$

and

(36)
$$P_{\infty }^{\mathrm{cl}}\left(q\right)=\left\{\begin{array}{cc} \frac{\beta^{\prime }\beta }{\beta^{\prime }+\beta }e^{-\beta q} & q\geq 0\\ \frac{\beta^{\prime }\beta }{\beta^{\prime }+\beta }e^{\beta^{\prime }q} & q<0 \end{array}\right.,$$

which are exactly the same as the long-time results obtained in Ref. [39].

### 3.3. Weak/Strong Coupling Limit in Finite Time

In the weak coupling limit 
$\kappa \ll \omega_{0}$
, the characteristic function of heat Equation (Equation 17) becomes

(37)
$$\begin{array}{cc} \chi_{\tau }^{w}\left(\nu \right) & =\frac{1}{\left(1+i\Xi \right)\left(1-i\Xi \Theta \right)\left(1-e^{-\kappa \tau }\right)+e^{-\kappa \tau }}. \end{array}$$


There is only one relaxation timescale associated to 
κ
. Such situation corresponds to the highly underdamped regime of the classical Brownian motion and a systematic method has been proposed to study the heat distribution [56], as well as the work distribution, under an external driving [82,83].

In the strong coupling limit 
$\kappa \gg \omega_{0}$
, the characteristic function of heat Equation (Equation 17) becomes

(38)
$$\begin{array}{cc} \chi_{\tau }^{s}\left(\nu \right)= & \frac{1}{\sqrt{\left(1+i\Xi \right)\left(1-i\Xi \Theta \right)\left(1-e^{-2\kappa \tau }\right)+e^{-2\kappa \tau }}}\\ \; & \times \frac{1}{\sqrt{\left(1+i\Xi \right)\left(1-i\Xi \Theta \right)\left(1-e^{-\frac{2\omega_{0}^{2}}{\kappa }\tau }\right)+e^{-\frac{2\omega_{0}^{2}}{\kappa }\tau }}}. \end{array}$$


The relaxation timescales of the momentum (the first factor) and the coordinate (the second factor) are separated. The long-time limits of both Equations (Equation 37) and (Equation 38) are equal to Equation (Equation 32). In classical thermodynamics, the usual overdamped approximation neglects the motion of the momentum; hence, the heat statistics derived under such an approximation is incomplete [52]. Actually, the momentum degree of freedom also contributes to the heat statistics.

### 3.4. Numerical Results

In Figure 1, we show the cumulative heat distribution function 
$\mathrm{Pr}\left(Q<q\right):=\int_{-\infty }^{q}P_{\tau }\left(q^{\prime }\right)dq^{\prime }$
 with different friction coefficients 
κ=0.01,1
 and 100, at the rescaled relaxation time 
$\tilde{\tau }=\kappa \tau =1$
 and 10. We set the mass 
$m_{0}=1$
 and the frequency 
$\omega_{0}=1$
 for the system harmonic oscillator, the inverse temperatures 
β=1
 and 
$\beta^{\prime }=2$
 for the initial equilibrium states of the heat bath and the system, respectively. The Planck constant is set to be 
ℏ=1,0.5,0.1
. With the decrease in *ℏ*, the quantum result Equation (Equation 17) approaches the classical result Equation (Equation 25). Thus, the quantum–classical correspondence of the heat distribution is demonstrated for generic values of the friction coefficient 
κ
.

For 
κ=0.01
 and 1, complete thermalization is achieved at 
$\tilde{\tau }=10$
. The left-lower and middle-lower subfigures show the identical distribution characterized by Equations (Equation 35) and (Equation 36). For 
κ=100
, the momentum degree of freedom is thermalized 
$\exp(-2\tilde{\tau })\approx 0$
 in Equation (Equation 38), while the coordinate degree of freedom remains frozen 
$\exp[-2\left(\omega_{0}^{2}/\kappa^{2}\right)\tilde{\tau }]\approx 1$
 in Equation (Equation 38). Thus, the distribution in the right-lower subfigure is different from the middle-lower subfigure.

In Figure 2, we illustrate the results of the mean value 
$\langle Q\rangle \left(\tau \right)$
 and the variance 
$\mathrm{Var}\left(Q\right)\left(\tau \right)$
 with different friction coefficients 
κ=0.01,1
 and 100. The parameters are the same as those in Figure 1. The quantum results approach the classical results with the decrease in *ℏ*. For 
κ=0.01
 and 1 (left and middle subfigures), complete thermalization is reached when 
$\tilde{\tau }>5$
. The mean value and the variance approach 
$\lim_{\tau \to \infty }\langle Q^{\mathrm{cl}}\rangle \left(\tau \right)=1/\beta -1/\beta^{\prime }$
 and 
$\lim_{\tau \to \infty }\mathrm{Var}\left(Q^{\mathrm{cl}}\right)\left(\tau \right)=1/\beta^{2}+1/\beta^{\prime 2}$
(gray horizontal lines). For 
κ=100
 (right subfigures), only the momentum degree of freedom is thermalized at this timescale. Thus, the mean value and the variance take half value of their long-time limits. When the coordinate degree of freedom is also thermalized in the long-time limit (
$\tilde{\tau }\gg \kappa^{2}/\omega_{0}^{2}=10^{4}$
), the mean value and the variance are expected to approach the same values as those in the middle subfigures.

## 4. Conclusions

Previously, the heat statistics of the relaxation processes has been studied analytically in open quantum systems described by the Lindblad master equation [39,40,50]. However, due to the rotating wave approximation and other approximations, such quantum systems do not possess a well-defined classical counterpart. Hence, the quantum–classical correspondence principle for heat distribution has not been well established.

In this paper, we study the heat statistics of the quantum Brownian motion model described by the Caldeira–Leggett Hamiltonian, in which the bath dynamics is explicitly considered. By employing the phase-space formulation approach, we obtain the analytical expressions of the characteristic function of heat at any relaxation time 
τ
 with an arbitrary friction coefficient 
κ
. The analytical results of heat statistics bring important insights to the studies of quantum thermodynamics. For example, in the classical limit, our results approach the heat statistics of the classical Brownian motion. Thus, the quantum–classical correspondence principle for heat statistics is verified in this model. Our analytical results provide justification for the definition of quantum fluctuating heat via two-point measurements.

We also discuss the characteristic function of heat in the long-time limit or with the extremely weak/strong coupling strength. In the long-time limit, the form of the characteristic function of heat reflects complete thermalization of the system. In addition, from the analytical expressions of the heat statistics, we can immediately verify the exchange fluctuation theorem of heat. The phase-space formulation can be further utilized to study the joint statistics of work and heat in a driven open quantum system, which would be beneficial in exploring the fluctuations of power and efficiency in finite-time quantum heat engines. 

## Figures and Tables

**Figure 1 entropy-23-01602-f001:**
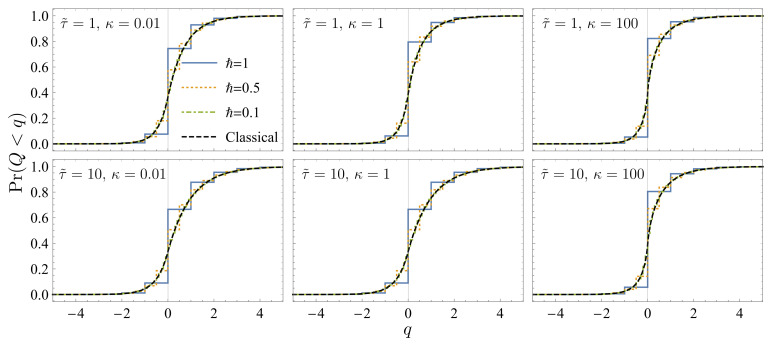
The cumulative heat distribution function 
Pr(Q<q)
. The choices of the parameters are given in the main text. We compare the results of the Caldeira–Leggett model (blue solid, orange dotted and green dot-dashed curves) in Equation (Equation 17) and those of the classical Brownian motion (black dashed curve) in Equation (Equation 25). The rescaled relaxation time is 
$\tilde{\tau }=\kappa \tau =1$
 in the upper subfigures and 
$\tilde{\tau }=10$
 in the lower subfigures. The left, middle and right subfigures illustrate the results for the weak (
κ=0.01
), intermediate (
κ=1
) and strong coupling strength (
κ=100
).

**Figure 2 entropy-23-01602-f002:**
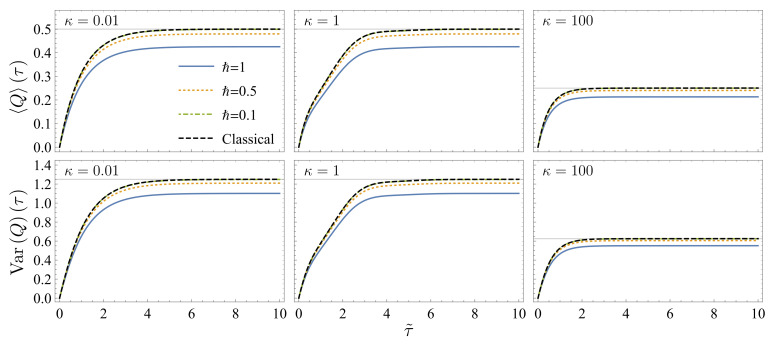
The evolution of the mean value 
$\langle Q\rangle \left(\tau \right)$
 (**upper subfigures**) and the variance 
$\mathrm{Var}\left(Q\right)\left(\tau \right)$
 (**lower subfigures**) of the heat statistics as functions of the rescaled time 
$\tilde{\tau }=\kappa \tau$
.

## Data Availability

Not applicable.

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
