# Peer review of "Quantum–Classical Correspondence Principle for Heat Distribution in Quantum Brownian Motion"

_entropy, 2021, doi:10.3390/e23121602_

Round 1

Reviewer 1 Report

The authors in this work explore the quantum to classical transition for the heat distribution of a quantum Brownian particle using path integrals. They show that the classical limit for the quantum heat distribution obtained using the two-point measurement scheme approaches correctly to the one expected from the classical Langevin equation. Moreover, they also show that the fluctuating heat at any time satisfies the exchange fluctuation theorem. The paper is very well written, and the authors have put in a substantial effort to shift the tedious mathematics to the appendices making the work accessible. I have a few comments that I would like the authors to consider before recommending this work for publication:

  1. I agree with the claims put forth by the authors, but I fail to see the novelty of this calculation. The quantum Langevin and the exact quantum master equation for the Brownian particle are well known [see Phys. Rev. E 55, 153 (1997)]. The quantum-classical correspondence is also well studied in these cases. Don’t these observations make the results of the current work trivial? A paragraph or so dedicated to explaining the novelty of this work would surely help the interested readers.
  2. In the introduction and later, the authors claim that heat statistics have been evaluated using a Lindblad master equation. This assessment is not true in general. The Redfield equation, which has lesser approximations than the Lindblad, has been used to evaluate the quantum heat [see Phys. Rev. B 85, 195452 (2012)] and its statistics [see J. Phys. Chem. C 114, 20362 (2010)].
  3. The exact Green’ function techniques [see Front. Phys. 9, 673 (2014)] have been used to evaluate the statistics of heat for the quantum Brownian particle. The approach uses the same set of assumptions as to the path integral formulation (like decoupled initial conditions with thermal states for system and baths). It is precisely equivalent to the two-point measurement scheme. This again raises the same question as point #1 about the novelty of this work. I also urge the authors to consider discussing the Green function methods as an alternative to path integrals.
  4. In the strong coupling regime, I believe the two-point measurement results for heat should coincide with Phys. Rev. B 92, 235440 (2015), but how does it differ from the definition arising via the Hamiltonian of mean force [Rev. Mod. Phys. 92, 041002 (2020)]. There is an ambiguity in the definitions of heat in the strong coupling regime since the system-bath interaction can be accounted for in different ways in the system’s internal energy. I believe that the authors’ definition of heat corresponds to the one used in Phys. Rev. B, but why is this the correct definition in the quantum regime? Moreover, observing a quantum-classical correspondence does not prove this since taking the limit of \hbar -> 0 inherently makes the Langevin equation classical, which can be obtained in the weak system-bath coupling limit. Could the authors comment on these issues that plague strong-coupling thermodynamics?
  5. Unfortunately, none of the citations were visible in the copy I reviewed. Hence, I cannot judge if the papers have been cited appropriately. I hope in the revised version this flaw is fixed. 

Overall, I do find the paper has its merits and should be published in some form. I urge the authors to consider the above points to make their article an interesting read even to the experts.

Author Response

See the PDF.

Reviewer 2 Report

In their interesting work  Jin-Fu Chen, Tian Qiu, and H.T. Quan  provide a deep study of the quantum heat statistics within the paradigmatic Caldeira-Leggett model. The authors use a precise phase-space formulation approach instead of a more standard Lindblad-like model that would involve a rotating wave aproximation.   With their approach they obtain rigorous analytical expressions for all times and for all the range of coupling parameters  (such as the friction coefficient). In particular the theory is analyzed in the long time limit  to understand thermal relaxation and  in order to study the strong and weak coupling regime.    This work is sound very interesting and should be published as it is.   

   I mention that the authors also provide a clean analysis of the hà0 limit regime  where they recover the   Classical ‘Brownian regime discussed  originally by Kramers.    Their definition of the classical regime is mathematical since we can no change the Planck constant; but of course for a physicist if would be better to consider  other alternative definitions of a classical limit, e.g. ,  by increasing the temperature. May be the authors could add a paragraph concerning this physical/philosophical issue. 

Round 2

Reviewer 1 Report

The authors in their revised version have addressed all my earlier concerns. Hence, I recommend this version for publication.